# State of the Art and Future Implications of SH003: Acting as a Therapeutic Anticancer Agent

**DOI:** 10.3390/cancers14041089

**Published:** 2022-02-21

**Authors:** Kangwook Lee, Bo-Young Youn, Yu-Jeong Choi, Seunghwan Moon, Jungkwun Im, Kyongha Cho, Seong-Gyu Ko, Chunhoo Cheon

**Affiliations:** 1Department of Preventive Medicine, Kyung Hee University, Seoul 02447, Korea; kwleeband@khu.ac.kr (K.L.); james_youn@khu.ac.kr (B.-Y.Y.); epiko@khu.ac.kr (S.-G.K.); 2Department of Science in Korean Medicine, Graduate School, Kyung Hee University, Seoul 02447, Korea; ehowlqk11@naver.com; 3Department of Global Public Health and Korean Medicine Management, Graduate School, Kyung Hee University, Seoul 02447, Korea; mshssang@gmail.com (S.M.); ijk0109@hanmail.net (J.I.); marquis99@naver.com (K.C.)

**Keywords:** anticancer agent, cancer, natural compound, phytochemical, herbal medicine

## Abstract

**Simple Summary:**

This review discusses the prominent highlights of SH003, a herbal mixture that has the potential to become a notable anticancer agent in the future. Although developing an anticancer drug may take a lengthy approval process for any natural compounds or herbal mixtures to validate the positive effects from both non-clinical and clinical studies, the previous studies of SH003 have so far shown positive results in various malignancies, from both non-clinical and clinical studies.

**Abstract:**

Cancer ranks as the first leading cause of death globally. Despite the various types of cancer treatments, negative aspects of the treatments, such as side effects and drug resistance, have been a continuous dilemma for patients. Thus, natural compounds and herbal medicines have earned profound interest as chemopreventive agents for reducing burden for patients. SH003, a novel herbal medicine containing Astragalus membranaceus, Angelica gigas, and Trichosanthes kirilowii, showed the potential to act as an anticancer agent in previous research studies. A narrative review was conducted to present the significant highlights of the total 15 SH003 studies from the past nine years. SH003 has shown positive results in both in vivo and vitro studies against various types of cancer cells; furthermore, the first clinical trial was performed to identify the maximum tolerated dose among solid cancer patients. So far, the potential of SH003 as a chemotherapeutic agent has been well-documented in research studies; continuous work on SH003’s efficacy and safety is required to facilitate better cancer patient care but is part of the knowledge needed to understand whether SH003 has the potential to become a pharmaceutical.

## 1. Introduction

Cancer is the leading cause of death worldwide, according to the World Health Organization [1]. The top three types of cancer in 2020 were breast cancer (2.26 million cases), lung cancer (2.21 million cases) and colon and rectum cancer (1.93 million cases). Furthermore, the estimated number of deaths is expected to reach nearly 13.2 million by 2030 [2]. It is unfortunate that cancer still remains one of the highest causes of death regardless of the substantial progress in cancer diagnosis and treatments.

There are many types of cancer treatments, including chemotherapy, radiotherapy, surgery, hormone therapy, immunotherapy, etc. [3]. A single or combination therapy can be applied depending on the type of cancer; among the therapies, chemotherapy is one of the most common treatments to kill cancer cells and to stop them from growing rapidly [4]. Despite the favor of chemotherapies, such therapies have led to numerous side effects, drug resistance and inadequate target specificity [4]. Thus, there has been a significant interest in finding natural anticancer agents. Developing natural-product-based drugs may take longer than traditional cancer drugs; natural-product-based drugs are known to overcome the harmful effects of chemotherapies and possess the strengths to target various cancer types. On the negative side, the quality control of the undiscovered active components and sources of natural compounds may be challenging.

As natural compounds, such as carotenoids, flavonoids, anthocyanins, or terpenoids, have shown positive aspects as anticancer agents, the concept of chemoprevention has continuously developed and also become an excellent source of anticancer drugs [5,6]. It is noteworthy that more than 60% of approved anti-tumorigenic drugs are derived from natural sources [7].

The rise of natural compounds as anticancer agents and its relation to cancer treatment has been discussed among patients since conventional chemotherapy has posed a considerable burden for cancer patients. It is important to note that several studies have shown the use of natural substances to reduce the toxic burden on a patient’s organs by a dose substitution with a natural compound, with a significantly positive effect [5]. Moreover, patients were able to tolerate high doses of natural compounds without any toxic effects [8].

Herbal medicines have also shown potential in reducing side effects while improving the immune system [9]. In particular, Chinese herbal medicine (CHM) has long been used to prevent and treat cancer in China. Huang et al. mentioned that arsenic trioxide, a toxic Chinese medicine, has been successfully applied in the clinical treatment of patients with acute promyelocytic leukemia; moreover, some formulae, including PHY906 based on Huang-Qin-Tang, have indicated a synergic effect with conventional drugs for improving the life quality of patients [10].

As previously stated, herbal medicines have been used with conventional chemotherapy or radiotherapy as a combination therapy to improve the efficacy of cancer treatment and reduce side effects, along with possible complications. Approximately between 28% and 98% of ethnic Chinese cancer patients in Asia have reported the utilization of integrative therapies with herbal medicines [11]; 25% to 47% of the patients living in North America also reported this.

With all that mentioned, natural compounds and herbal medicines, as well as herbal mixtures, have shown a considerable number of positive effects and have offered a superb opportunity for discovering therapeutic agents for the treatment of cancer.

Thus, the study aims to highlight the potentials of SH003, a herbal mixture, working as a vital anticancer agent, while providing evidence that it targets multiple metabolic pathways in both single-dose therapy and combined therapy along with conventional chemotherapeutics.

## 2. Methods and Materials

This study was written as a form of narrative review. All articles related to SH003 were searched for via PubMed with the keyword “SH003” in the textbox. There was a total number of 16 articles identified, and a protocol paper was excluded; therefore, the remaining 15 articles were carefully examined and summarized.

## 3. Characteristics of SH003

SH003 is a mixture of Huang-Qi (Astragalus membranaceus; AG), Dang-Gui (Angelica gigas; AM), and Gua-Lou-Gen (Trichosanthes Kirilowii; TK), which are traditionally used in East Asian medicine. According to the theory of traditional medicine, the effect of Huang-Qi is to tonify qi, the effect of Dang-Gui is to tonify blood, and the effect of Gua-Lou-Gen is to disperse swelling and expel pus [12]. SH003 extracts were provided by HANPOONG (HANPOONG PHARM & FOODS Co., Jeonju, Korea), which followed good manufacturing practice (GMP) procedures. In brief, Astragalus membranaceus (333 g), Angelica gigas (333 g), and Trichosanthes kirilowii Maximowicz (333 g) were mixed at a 1:1:1 ratio and then extracted with 10 times the volume of 30% ethanol at 100 °C for 3 h. This process was performed 2 times. The extract was dried at reduced pressure (40 Torr) at 60 °C for 18 h. Notably, the experimental study proved that Danggwibohyeoltang, a mixture of AM and AG, inhibits the immune-enhancing effect [13]. Meanwhile, we demonstrated that the ethanol extracts of TK induce apoptosis through inhibition of STAT3 activity in triple-negative breast cancer MDA-MB-231 cell lines [14]. Taken together, we finally decided to develop a novel anti-cancer herbal mixture by combining AG, AM and TK [15]. As shown in Figure 1, the anti-cancer effect of SH003 has been demonstrated by several publications.

Toxicity studies with GLP regulations showed that SH003 is safe in rats [16]. In brief, rats were orally administrated SH003 (0, 500, 1000 and 2000 mg/kg) every day for 2, 4 and 13 weeks. After administration, body weight, mortality, food intake, clinical signs, hematological values, serum biochemical values, relative organ weights and histopathology were recorded. The results show that the oral administration of SH003 does not result in any toxicological phenotypes. A dose-escalation study revealed that the no-observed-adverse-effect level (NOAEL) is higher than 2500 mg/kg for both male and female rats. Furthermore, we further investigated the herb–drug interaction. Human liver microsomes were incubated with SH003, and then substrates including phenacetin, coumarin, paclitaxel, diclofenac, (±)-mephenytoin, dextromethorphan and midazolam were added. As a result, SH003 exhibited a minimal inhibitory effect on all CYP isozymes, suggesting that there was no herb–drug interaction. Based on the findings, we concluded that SH003 might be safe for cancer patients, and further clinical studies should be carried out to confirm the exact benefit of SH003 in cancer therapy.

In recent years, clinical trials to confirm safety have also been conducted. A phase I study evaluating the maximum tolerated dose for SH003 administration alone confirmed the daily safe dose to be up to 4800 mg [17]. A clinical trial to ensure the maximum tolerated dose when administered in combination with docetaxel is also being conducted [18].

## 4. Current Advances of SH003 in Tumor Suppression

Herbal medicines have been used to prevent or inhibit tumor growth and metastasis. SH003 plays a crucial role in regulating various types of cancer (Figure 2 and Table 1). In this regard, the anti-cancer effect and mechanisms of SH003 against a wide range of cancers based on in vitro and in vivo studies are discussed here.

### 4.1. Breast Cancer

Breast cancer is a leading cause of cancer death in women [19]. Based on the immunohistochemistry biomarkers, including estrogen receptor (ER), progesterone receptor (PR), and HER2, breast cancer is classified into the following subgroups; luminal A (ER+, PR±, HER2−), luminal B (ER+, PR±, HER2+), HER2 (ER−, PR−, HER2 overexpression) and basal-like triple-negative (ER−, PR−, HER2−) breast cancer [20,21]. While conventional targeted therapies on ER, PR, or HER2 have been applied for breast cancer patients with luminal A, luminal B, or HER2 subtypes, the treatment options for triple-negative breast cancer (TNBC) are still limited, resulting in poor prognosis [22,23]. Therefore, the discovery of new anti-cancer agents for the treatment of TNBC patients is still needed.

In 2014, our group reported the tumor-suppressive effect of SH003 on triple-negative breast cancer [15]. It was shown that SH003 inhibits tumor growth and metastasis to the lung in the mouse xenograft model via the down-regulation of vascular endothelial cell marker (CD31). From in vitro results, SH003 inhibited the growth of various breast cancer cell lines, including luminal A, luminal B, HER2, and TNBC subgroups, when compared with the normal epithelial cell. Moreover, treatment with SH003 inhibited migration, invasion, and the anchorage-dependent colony formation of MDA-MB-231 TNBC cell lines. Western blot analysis revealed that SH003 decreased the expression of STAT3 phosphorylation and STAT3-dependent proteins. Meanwhile, SH003 also blocked the nuclear translocation of phosphorylation and the transcriptional activities of STAT3 in MDA-MB-231 cells. By inhibiting STAT3 activation, SH003 decreased the production of STAT3-mediated IL-6. Evidence showed for the first time that SH003 could be a novel anti-cancer herbal mixture for TNBC by inhibiting the STAT3-IL-6 autocrine loop. Another study was performed to define the growth-inhibitory effect of SH003 on p53-mutant TNBC [24]. SH003 has a significant anti-cancer effect via p73-mediated apoptosis in TNBC cells with p53 mutation. In 2017, Choi et al. found that SH003 suppresses the growth of TNBC cell lines by inducing autophagy and apoptosis [16]. SH003 induces autophagy by inhibiting STAT3 and mTOR signaling pathways, inducing lysosomal p62/SQSTM1 accumulation-mediated reactive oxygen species (ROS) generation. SH003-induced p62 accumulation in autophagosome caused ROS-mediated apoptotic death in TNBC cell lines. In a mouse xenograft model, SH003 dose-dependently inhibited tumor growth with the down-regulation of Ki-67 and phosphorylation of STAT3, and up-regulation of p62. Accumulating data from several studies about SH003 demonstrate that SH003 is a novel herbal mixture for TNBC treatment by causing apoptotic cell death and autophagy.

Conventional chemotherapy has faced limitations such as multi-drug resistance [25,26]. To overcome the resistance, it is necessary to develop an alternative therapeutic strategy, such as combination chemotherapy, which sensitizes cancer cells to each drug, and decreases side effects by reducing the dose [27,28]. In recent years, several anti-cancer agents, including taxane and doxorubicin, have been trialled in combinational therapy against breast cancer. Moreover, a combination of herbal medicines and conventional chemotherapies has been suggested [28,29]. Considering a combinational strategy for TNBC treatment, we further investigated whether SH003 and doxorubicin exhibit a synergistic effect on TNBC treatment [30]. SH003 and doxorubicin synergistically inhibited the cell viability of TNBC MDA-MB-231 cell lines. Moreover, this synergistic cell death was associated with caspase-dependent apoptosis in MDA-MB-231 cell lines. The tumor growth of MDA-MB-231 in mouse xenograft models was synergistically decreased by the combinational treatment of SH003 with doxorubicin. Thus, this study suggests that a combination treatment of SH003 and doxorubicin would be a novel strategy for TNBC treatment. In addition, further studies examined whether SH003 treatment can overcome paclitaxel resistance in ER+ breast cancer [31]. Choi et al. investigated the inhibitory effect of SH003 on MDR1 activity in paclitaxel-resistant ER+ breast cancer MCF7 cell lines (MCF7/PAX). The results show that SH003 and paclitaxel inhibited the viability of MCF7/PAX in a synergistic manner. Of note, the SH003 inhibition of MDR1 expression sensitized MCF7/PAX to paclitaxel. Seo et al. further examined the molecular mechanism of SH003 in overcoming paclitaxel resistance in MCF7/PAX. SH003 decreased the viability of MCF7/PAX by inducing cell cycle arrest and apoptosis [32]. Moreover, SH003 treatment down-regulated the expression of phosphor-STAT3 and prevented the translocation of STAT3 into the nucleus. Taken together, the above studies demonstrated that SH003 could overcome drug resistance by targeting MDR1 and STAT3 and suggested that SH003 is likely to be a partner with conventional chemotherapy for overcoming drug resistance.

### 4.2. Lung Cancer

According to the American Cancer Society, lung cancer has been by far the top-ranked cancer for death among both men and women, from 1998 to 2021 [19,33]. According to microscopic features, lung cancer is classified into non-small cell lung cancer (NSCLC) and small-cell lung cancer (SCLC) [34,35]. NSCLC patients commonly receive platinum or taxane-based regimens or targeted therapy for epidermal growth factor receptor (EGFR) [36,37]. Docetaxel—taxane with anti-mitotic properties—is an effective anti-cancer agent, causing cell cycle arrest and apoptosis in NSCLC [38,39,40]. However, docetaxel-mediated chemoresistance and severe side effects, including peripheral neuropathy, anorexia, and cachexia, are still the cause of treatment failure in cancer patients [41,42]. Recent studies have focused on the development of novel treatment strategies by combining chemotherapy with herbal medicines for NSCLC treatment [29,43]. Several clinical studies demonstrated the survival benefit of chemotherapy in combination with traditional Chinese herbal medicines in cancer patients [44,45,46]. Thus, we hypothesized that the combination treatment of SH003 and docetaxel is effective for NSCLC patients [47]. The results of an MTT cell viability assay showed that the co-treatment of SH003 and docetaxel synergistically inhibited the viability of NSCLC A549 and H460 cell lines. Furthermore, FACS and Western blot analysis demonstrated that combination treatment with the lowest combination index effectively induced apoptotic cell death when compared with a single treatment with SH003 or docetaxel. Moreover, since EGFR has a pivotal role in cancer progression, metastasis, and drug resistance in NSCLC, the combinational effect of SH003 and docetaxel on the EGFR signaling pathway was also examined. The results show that this combination treatment synergistically inhibited the expression of phosphor-EGFR (Y1068) in H460 cell lines with or without epidermal growth factor (EGF). Additionally, downstream of the EGFR, the JAK/STAT3 signaling pathway was decreased under combination treatment. An in vivo study confirmed that the combination of SH003 and docetaxel more strongly inhibited the tumor growth than single treatment with each agent. Immunohistochemical staining showed that combination treatment effectively increased the apoptosis marker (cleaved caspase 3) and decreased survival marker (Ki-67), phospho-EGFR(Y1068), and phospho-STAT3 (Y705). Thus, this study concluded that docetaxel treated with SH003 synergistically induced apoptotic cell death by inhibiting the EGFR–STAT3 signaling pathway in EGFR wild-type NSCLC.

### 4.3. Other Malignancies

Besides breast and lung cancer, the anti-cancer effects of SH003 on other cancer types have been investigated by non-clinical studies [48,49,50]. In 2016, Choi et al. investigated the anti-cancer effects of SH003 in prostate cancer cells [48]. SH003 treatment dose-dependently inhibited the viability of prostate cancer DU145 cell lines by inducing apoptosis. Moreover, SH003 induced apoptotic cell death via inhibiting the ERK signaling pathway, while ERK overexpression reversed it. In the case of cervical cancer, Lee et al. demonstrated that SH003 exhibits an anti-cancer effect by regulating cell cycle arrest and apoptotic cell death [49]. Kim et al. reported the effect of SH003 on the autophagic death of gastric cancer cells [50]. In gastric cancer, SH003 treatment dose- and time-dependently inhibited the viability of various gastric cancer cell lines, the inhibition of which was associated with the induction of apoptosis. Meanwhile, SH003 also induced ER stress via PERK-ATF4-CHOP signaling, inhibiting G9a by suppressing STAT3 phosphorylation and activating autophagy. Of note, SH003-induced ER stress induced BNIP3-related autophagic death via the suppression of STAT3/G9a axis under hypoxic conditions.

### 4.4. Tumor Angiogenesis

Tumor angiogenesis is crucial for tumor growth and distant metastasis [51,52]. The inhibition of tumor angiogenesis has been considered a potential target for cancer treatment. Vascular endothelial cell growth factor (VEGF) released from cancer cells binds to VEGF receptor (VEGFR) on vascular endothelial cells, resulting in neo-angiogenesis. Based on the finding that SH003 suppressed TNBC tumor growth with the down-regulation of endothelial cell marker (CD31) in a mouse xenograft model [15], Choi et al. performed a further study to prove an anti-angiogenic effect of SH003 [53]. While VEGF induced the migration, invasion, and tube formation of human vascular endothelial cells (HUVEC), SH003 treatment inhibited it. Moreover, SH003 dose- and time-dependently inhibited the VEGF-mediated activation of VEGFR2 downstream in HUVEC. In the mouse xenograft model, a low dose of SH003 (2 mg/kg) decreased tumor growth. Immunohistochemistry results show that SH003 reduced the expression of Ki-67, phospho-VEGFR2 and vascular endothelial cell marker (CD31), and increased apoptosis marker (cleaved caspase 3) in tumor tissues, suggesting that SH003 administration decreased the tumor growth by inhibiting tumor angiogenesis. Further vascular leakage assays confirmed the anti-angiogenic effect of SH003. Thus, these data encourage SH003 development for a novel anti-angiogenic agent, as well as a cytotoxic agent.

### 4.5. Managing Cancer-Related Adverse Effect

#### 4.5.1. Chemotherapy-Induced Peripheral Neuropathy

As mentioned above, chemotherapy causes severe side effects. Chemotherapy-induced peripheral neuropathy (CIPN) is one of the painful side effects characterized by damage to peripheral neurons [54]. Cancer patients who receive docetaxel experience acute pain syndrome [55,56]. Regardless of the therapeutic benefit of docetaxel, patients may consider the cessation of cancer treatment because of painful symptoms. Unfortunately, there have been no effective options for the treatment of CIPN. Researchers reported that herbal medicines could be novel therapeutic options for relieving cancer-related side effects, including CIPN [29,57]. In 2021, the therapeutic effect of the novel herbal mixture SH003 on docetaxel-induced neuropathic pain was reported [58]. Lee et al. demonstrated that SH003 alleviated mechanical allodynia in the docetaxel-induced mouse CIPN model. Intravenous docetaxel injection induced the degeneration of intraepidermal nerve fibers in the feet of C57BL/6 mice, but SH003 treatment alleviated it. Additionally, SH003 decreased the upregulation of TNF- α and IL-6 in plasma and increased expression of phospho-NF-κB and phospho-STAT in L4-L6 spinal cord and sciatic nerves in docetaxel-injected mice. Based on these findings, therapeutic indications of SH003 can be expanded to CIPN in addition to killing cancer.

#### 4.5.2. Immune-Enhancing Effect

The immune system of cancer patients who receive several therapies, including chemotherapy and radiotherapy, is commonly weakened, resulting in tumor progression and poor prognosis [59,60]. Several studies reported that herbal medicines and their derivatives exhibit immunostimulatory effects [61,62]. Han et al. demonstrated that SH003 improves immunosuppression via the activation of immune cells such as macrophages, splenocytes, and NK cells [63]. SH003 treatment increased the production of colony-stimulating factors, IL-2, IL-6, IL-12, TNF-α, nitric oxide, and iNOS. Moreover, the transcription factor NF-κB was enhanced by SH003. In splenocytes, SH003 also stimulated the production of IFN-γ, IL-2, IL-12, TNF-α, and nitric oxide. The splenic lymphocyte proliferation and splenic NK cell activity were increased by SH003 treatment. In the cyclophosphamide-induced immunosuppression murine model, SH003 alleviated immunosuppression with the increased production of IFN-γ, IL-2, IL-6, IL-12, and TNF-α in serum and spleen. These data suggest that SH003 could be applied as an immunostimulatory agent for immunosuppressive disease.

### 4.6. Anti-Cancer Effect of SH003 Derivatives

Since SH003 is a herbal mixture that contains multiple phytochemicals, it was necessary to decipher what compounds of SH003 show anti-cancer effects. From 2012 to 2020, the SH003 research group has found apigenin, cucurbitacin D, decursin, kaempferol, and quercetin as potential anti-cancer agents (Table 2). In brief, a number of studies have demonstrated that each putative active compound mainly regulates the signaling pathways in apoptosis or autophagy, which are the key anticancer targets of SH003. Therefore, SH003 is expected to have anticancer effects through the synergistic effect of these compounds, although non-clinical studies should prove this. Moreover, the previous research focused on the chemical profiling of SH003 identified several constituents, whereas the anticancer effect of single components and their combination are still unknown [16]. However, there are still undiscovered active components in SH003. Thus, further studies should be performed to identify new compounds in SH003 and to investigate the synergistic interactions of multiple components.

#### 4.6.1. Apigenin

One of the phytoestrogens, apigenin induced the cell p53-dependent apoptotic death of HER2-overexpressing breast cancer MCF7 cells engineered to overexpress oncogenic HER2 (MCF7-HER2) [64]. Moreover, apigenin-induced cell death resulted from the inhibition of STAT3 and the NF-κB signaling pathway. Another study demonstrated that apigenin inhibits hypoxia-induced VEGF production from HER2-overexpressing breast cancer MDA-MB-453 cells [65]. Furthermore, apigenin decreased MDR1 expression in doxorubicin-resistant MCF7 cells by blocking the STAT3 signaling pathway, which could contribute to overcoming multi-drug resistance [66].

#### 4.6.2. Cucurbitacin D

Cucurbitacin D has been known to display anti-cancer and anti-inflammatory activity [67,68]. In 2015, Ku et al. demonstrated that cucurbitacin D kills doxorubicin-resistant MCF7 cells by inducing cell cycle arrest and apoptosis [69]. Moreover, cucurbitacin D-induced cell death was associated with inhibiting STAT3 and the NF-κB signaling pathway. It was also reported that Tk or cucurbitacin D combined with cisplatin/pemetrexed synergistically induces the apoptotic death of non-small-cell lung cancer H1299 cells by suppressing ErbB3 signaling pathways [70]. In 2020, Hong et al. proved that cucurbitacin D blocks EGF binding to EGFR, followed by the inhibition of growth and migration of gefitinib-resistant NSCLC cells. This study concluded that cucurbitacin D could be a novel agent for overcoming gefitinib resistance via targeting the EGF–EGFR signaling pathway in gefitinib-resistant NSCLC. In recent years, the anti-cancer effect of cucurbitacin D against pancreatic cancer was reported [71]. Cucurbitacin D dose-dependently inhibited the viability of pancreatic cancer cell lines by inducing G2/M cell cycle arrest and apoptosis. Moreover, cucurbitacin D-mediated ROS production regulates cell cycle arrest and apoptosis in pancreatic cancer cell lines. It is worth noting that cucurbitacin D-induced ROS generation sequentially activates the p38/c-jun signaling pathway and, in turn, triggers apoptotic cell death.

#### 4.6.3. Decursin

Decursin has been known as a major component of AG, which exhibits a potential anti-cancer effect on several malignancies [72]. In 2016, Choi et al. reported that the main active component of AG, decursin, sensitizes doxorubicin-resistant ovarian cancer cells to doxorubicin [73]. Co-treatment of decursin and doxorubicin synergistically induces cell death and apoptotic cell death. Moreover, it is worth noting that overcoming doxorubicin resistance by decursin treatment is mediated by the inhibition of P-glycoprotein expression.

#### 4.6.4. Kaempferol

Kaempferol is a naturally occurring flavonoid and possesses anti-cancer activity against several malignancies [72]. In 2018, Kim et al. reported kaempferol’s anti-cancer effect and molecular mechanism on gastric cancer [73]. Kaemferol induced autophagic cell death by inhibiting p62 and activating IRE1-JNK-CHOP signaling. Moreover, kaempferol epigenetically modified G9a (HDAC/G9a) followed by autophagic cell death.

#### 4.6.5. Quercetin

Quercetin has been shown to exhibit antioxidant, anti-inflammatory, and anti-angiogenic properties [74,75,76]. Notably, the anti-cancer effect of quercetin has been well-documented in accumulating numbers of publications, both in cell and animal models [77,78]. In 2016, our group reported the anti-proliferative effects of quercetin on HER2-overexpressed breast cancer BT-474 cells [79]. The results show that quercetin dose- and time-dependently suppresses the growth of BT-474 cells. According to the anchorage-dependent and -independent assay, quercetin effectively inhibited the colony formation of BT-474 cells. Moreover, quercetin treatment triggered G1-phase arrest of cell cycle progression and caspase-dependent extrinsic apoptosis, but not intrinsic apoptosis. Quercetin-induced apoptotic cell death was accompanied by a decrease in STAT3 expression and transcriptional activity.

## 5. A Rapid Review of the Clinical Trials of SH003

Current clinical trials of SH003 in development are listed in Table 3. This first-in-human phase 1 study of SH003 was conducted with patients with solid cancers; the study may be the first phase 1 clinical trial of a herbal mixture conducted in Korea. The patients were recruited and administered one to four tablets of SH003 for three weeks at the designated dose level. Patients in cohorts 1, 2, and 3 took 1200, 2400, or 4800 mg/day in this respect. The aim of this clinical trial was to examine the maximum tolerated dose (MTD) of SH003 alone. The MTD is defined as the maximum dose that does not cause adverse events of grade 3 or more according to the Common Terminology Criteria for Adverse Events (CTCAE) version 4.03 from the National Cancer Institute in more than two of six participants. Since no adverse events of grade 3 or more were observed, 4800 mg/day was the highest dose determined to be the MTD of SH003. Considering the efficacy of SH003, a phase 2 clinical trial with patients with cancer at a dose of 4800 mg/day ought to be planned.

As previously mentioned, combination therapy with conventional therapies has earned great interest in treating cancer patients. As SH003 has been investigated for a feasible combination with conventional drugs, a clinical trial has been ongoing, confirming the safety of SH003 and docetaxel in patients with solid cancer.

## 6. Conclusions and Future Perspectives

Although natural compounds and extracts of herbal medicines are promising chemopreventive agents, there is still a lack of evidence. However, SH003 so far has been proved to possess positive effects, showing greater tumorigenicity and cytotoxic effects while inhibiting tumor growth and suppressing the proliferation of cells in various targeted lung, breast, prostate, cervical, and gastric cancer cells. The next step for SH003 is to perform enough RCTs to analyze patients’ performance status, tolerance of side effects, level of drug-related toxicity, tumor response, and length of survival to improve the efficacy and safety. Furthermore, combination therapies with conventional chemotherapy or radiotherapy will become exceptionally valuable in the future, such that more research efforts with SH003 in the field of integrative cancer therapies are much needed. Well-designed RCTs will provide the most accurate estimates of treatment effects while helping physicians to plan the best treatment course for patients. Overall, the potential of SH003 as a chemotherapeutic agent has been well-presented so far; however, continuous further work is warranted to facilitate better cancer patient care in the near future.

## Figures and Tables

**Figure 1 cancers-14-01089-f001:**
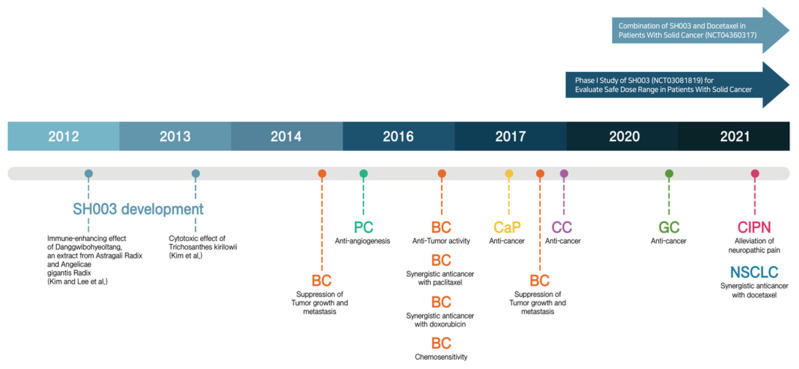
The timeline of SH003 (BC: breast cancer; PC: pancreatic cancer; CaP: prostate cancer; CC: cervical cancer; GC: gastric cancer, CIPN: chemotherapy-induced peripheral neuropathy and NSCLC: non-small cell lung cancer).

**Figure 2 cancers-14-01089-f002:**
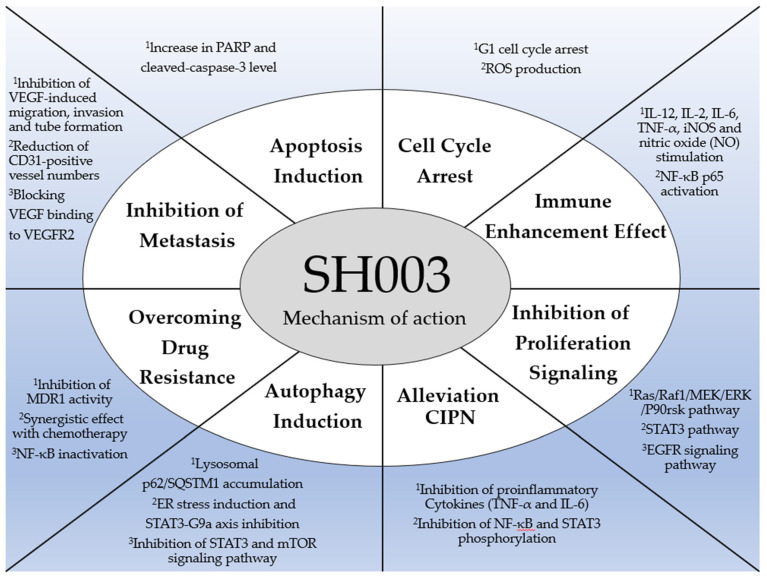
A mechanistic review of SH003.

**Table 1 cancers-14-01089-t001:** A summary of the effects of SH003 on cancer, immune system and chemotherapy-related side effects.

Cancer Type	Cell Type	Proposed Effects	Methods	Mechanism	Refs.
Breast cancer	MDA-MB-231	Suppression of tumor growth and metastasis	in vitro (0–500 μg/mL) in vivo (500 mg/kg)	Inhibition STAT3-IL-6 Signaling	[15]
	MDA-MB-231 and HCC-38	Pro-apoptosis and autophagy induction	in vitro (0–500 μg/mL) in vivo (10, 100, 500 mg/kg)	Accumulation p62 in autolysosomes	[16]
	Hs578T, MDA-MB-231, ZR-75-1, MCF7 and T47D	Pro-apoptosis, synergistic anticancer effect with paclitaxel	in vitro (0–200 μg/mL)	Increase in p73 expression	[24]
	MDA-MB-231	Pro-apoptosis, synergistic anticancer effect with doxorubicin	in vitro (0–500 μg/mL) in vivo (500 mg/kg)	Caspase cascade activation	[30]
	Paclitaxel-resistant breast cancer cell (MCF-7/PAX)	Overcoming drug resistance	in vitro (0–500 μg/mL)	Inhibition of MDR1 activity, inhibition of STAT3 signaling pathway	[31,32]
Endothelial cells	Human umbilical vein endothelial cells (HUVECs)	Anti-angiogenesis	in vitro (0–50 μg/mL) in vivo (2 mg/kg)	Blockade VEGF binding to VEGFR2	[53]
Prostate cancer	DU145	Pro-apoptosis	in vitro (0–500 μg/mL)	Inhibition ERK signaling pathway	[48]
Cervical cancer	HeLa	Pro-apoptosis	in vitro (0–500 μg/mL)	G1 cell cycle arrest, ROS generation	[49]
Gastric cancer	AGS and SNU-638	Autophagic cell death	in vitro (0–400 μg/mL)	ER stress induction and inhibition of STAT3-G9a axis	[50]
Non-Small Cell Lung Cancer	H460	Synergistic anticancer effect with docetaxel	in vitro (0–500 μg/mL) in vivo (557.569 mg/kg)	Inhibition EGFR–STAT3 signaling pathway	[47]
C57BL/6 Mice	Docetaxel-Induced Neuropathy Mouse Model	Alleviation of docetaxel-induced neuropathic pain	in vivo (557.569 mg/kg)	Inhibition of proinflammatory cytokines (TNF-α and IL-6), NF-κB and STAT3	[58]
Immune cell	Macrophage (RAW 264.7) and NK cell	Immune-enhancing activity	in vitro (0–500 μg/mL) in vivo (400 mg/kg)	Production immunostimulatory cytokines and NO, activation of NF-κB	[63]

**Table 2 cancers-14-01089-t002:** A summary of SH003 derivative-induced effects on cancer treatment.

Herb	Active Compound	Cancer Type/Cell Type	Mechanism	Refs.
Astragalus membranaceus, Trichosanthes Kirilowii Maxim.	Apigenin (0–40 μM [64]) (0–100 μM [65,66])	Breast cancer (MCF-7. SK-BR-3, BT-474, MDA-MB-453, MCF-7 HER-2 and MCF7/ADR)	Inhibition of STAT3 and NFκB signaling, downregulation of MDR1 expression	[64,65,66]
Astragalus membranaceus, Trichosanthes Kirilowii Maxim.	Quercetin (0–100 μM)	Breast cancer (BT-474)	Apoptosis through inhibition of STAT3	[79]
Astragalus membranaceus	Kaempferol (0–100 μM)	Gastric cancer (AGS, SNU-216, NCI-N87, SNU-638, and MKN-74)	Activaiton of IRE1-JNK-CHOP pathway, G9a inhibition	[73]
Trichosanthes Kirilowii Maxim.	Cucurbitacin D (0–2 μg/mL [69]) (0–10 μM [70]) (0–0.8 μM [71])	Doxorubicin-resistant human breast carcinoma (MCF7/ADR)	Inhibition of STAT3 and NFκB signaling	[69]
		Non-small-cell lung cancer (H1299, HCC827 and HCC827GR)	ErbB3 and EGFR signaling inhibition, synergistic effect with CDDP/PXD, overcoming gefitinib resistance	[70]
		Pancreatic cancer (Capan-1)	G2/M phase arrest through ROS-p38 pathway	[71]
Angelica gigas Nakai	Decursin (0–50 μg/mL)	Doxorubicin-resistant human breast carcinoma (MCF7/ADR)	Inhibition of P-glycoprotein expression	[73]

**Table 3 cancers-14-01089-t003:** Current clinical trials of SH003 in development.

Clinical Trial	Phase	Study Description	Intervention	Targets	Sponsors and Collaborators	Refs.
NCT03081819 (ClinicalTrials.gov)	Phase I	SH003 for evaluating safe dose range in patients with solid cancer	SH003	Solid tumor, adult	Kyung Hee University Medical Center	[17]
NCT04360317 (ClinicalTrials.gov)	Phase I	Safety of the combination of SH003 and docetaxel in patients with solid cancer	Combination of SH003 and docetaxel	Solid tumor	Kyung Hee University Medical Center	[18]

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
