# Peer review of "State of the Art and Future Implications of SH003: Acting as a Therapeutic Anticancer Agent"

_cancers, 2022, doi:10.3390/cancers14041089_

Round 1
Reviewer 1 Report
State-of-art and future implications of SH003: Acting as a therapeutic anticancer agent
The review systematically describes the recent literature regarding the activity of SH003, a herbal mixture, as a potential chemotherapeutic and adjuvant for cancer treatment.
Here are presented a series of suggestions and small concerns made to improve the clarity and readability of the presented review.
The simple summary needs to be more clear and concise. The authors mention that natural compounds and herbal mixtures might take an exceptionally long development and approval process to validate their good aspects as anticancer agents. It is not clear what they refer to as a “long development” and the validation of “good aspects”. Please also be more precise in what the authors refer to as “positive results” from the previous literature.
Please improve the last sentence of the abstract. Conducting studies on a compound’s efficacy and safety does not only “facilitate patient care” but is part of the knowledge required to understand if a chemical has the potential to become a pharmaceutical.
Please consider rephrasing the sentence on the first page, lines 44-45. Aside from mentioning the drawbacks of chemotherapy, the authors could discuss the advantages and problems of natural product-based drugs and their benefits to traditional cancer treatments.
There is a problem with how the term “Chemopreventive agent” in the introduction is used. Chemopreventive agents impede cancer development through mechanisms usually related to maintaining DNA integrity. Because the activity of chemopreventive agents focuses on avoiding the events that lead to cancer initiation, they are especially relevant for healthy individuals in whom cancer may never develop. The information in this review supports the activity of SH003 as an anticancer drug and a chemotherapy adjuvant when used to treat cancer. Considering this, please improve the text on page 2, lines 49-80.
Figure 2 could be improved to be more informative; more than the mechanism of action, the figure represents aspects of the anticancer activity reported for SH003.
Please include citations to the phrases where the authors use the statement “We”, wherever it is not included already (e.g., page 2, lines 95-96; page 3, lines 105, 109; …etc.).
Please add a reference to the sentence on page 5, lines 154-156.
Please improve the text in section 6. Conclusions and future perspectives, page 11, lines 375-387; this text summarizes the rationale of this review, and it should be clear and convincing. Please be clear on what “positive effects” reflect for SH003 activity (line 377), there is an extra apostrophe in the word patients’ (line 379), the sentence in lines 380-383 needs to be rewritten to assure its clarity, and the finishing sentence seems incomplete.
Author Response
Dear Reviewer:
First of all, we would like to express our deepest gratitude for reviewing our manuscript. Please review the answers below to your each comment. Additionally, the changes will be visible in red from the revised manuscript.
The simple summary needs to be more clear and concise. The authors mention that natural compounds and herbal mixtures might take an exceptionally long development and approval process to validate their good aspects as anticancer agents. It is not clear what they refer to as a “long development” and the validation of “good aspects”. Please also be more precise in what the authors refer to as “positive results” from the previous literature.
=] Although developing an anticancer drug may take a lengthy approval process for any natural compounds or herbal mixtures to validate positive effects from both non-clinical and clinical studies, the previous studies of SH003 have so far shown positive results in various malignancies from both non-clinical and clinical studies.
Please improve the last sentence of the abstract. Conducting studies on a compound’s efficacy and safety does not only “facilitate patient care” but is part of the knowledge required to understand if a chemical has the potential to become a pharmaceutical.
=] So far, the potential of SH003 being a chemotherapeutic agent has been well-documented in research studies; continuous work of providing SH003's efficacy and safety is required to facilitate better cancer patient care but is part of the knowledge needed to understand whether a SH003 has the potential to become a pharmaceutical.
Please consider rephrasing the sentence on the first page, lines 44-45. Aside from mentioning the drawbacks of chemotherapy, the authors could discuss the advantages and problems of natural product-based drugs and their benefits to traditional cancer treatments.
=] Thus, there has been a significant interest in finding natural anticancer agents. Developing natural product-based drugs may take longer than traditional cancer drugs; natural product-based drugs are known to overcome the harmful effects of chemotherapies and possess the strengths to target various cancer types. On the negative side, quality control of the undiscovered active components and sources of natural compounds may be challenging.
There is a problem with how the term “Chemopreventive agent” in the introduction is used. Chemopreventive agents impede cancer development through mechanisms usually related to maintaining DNA integrity. Because the activity of chemopreventive agents focuses on avoiding the events that lead to cancer initiation, they are especially relevant for healthy individuals in whom cancer may never develop. The information in this review supports the activity of SH003 as an anticancer drug and a chemotherapy adjuvant when used to treat cancer. Considering this, please improve the text on page 2, lines 49-80.
=] Changed the chemopreventive word to anticancer
Figure 2 could be improved to be more informative; more than the mechanism of action, the figure represents aspects of the anticancer activity reported for SH003.
=] Please review our updated figure in our revised manuscript.
Please include citations to the phrases where the authors use the statement “We”, wherever it is not included already (e.g., page 2, lines 95-96; page 3, lines 105, 109; …etc.).
=] The paragraph is related to the reference number 16.
Please add a reference to the sentence on page 5, lines 154-156.
=] The paragraph is related to the reference number 16.
Please improve the text in section 6. Conclusions and future perspectives, page 11, lines 375-387; this text summarizes the rationale of this review, and it should be clear and convincing. Please be clear on what “positive effects” reflect for SH003 activity (line 377), there is an extra apostrophe in the word patients’ (line 379), the sentence in lines 380-383 needs to be rewritten to assure its clarity, and the finishing sentence seems incomplete.
=] Although natural compounds and extracts of herbal medicines are promising chemopreventive agents, there is still a lack of enough evidence. However, SH003 so far has proved the positive effects that shows greater tumorigenicity and cytotoxic effects while inhibiting tumor growth and suppressing proliferation of cells in various targeted lung, breast, prostate, cervical, gastric cancer cells. The next step of SH003 is to perform enough RCTs to analyze patients’ performance status, tolerance in side effects, level of drug-related toxicity, tumor response, and length of survival to improve the efficacy and safety. Further, combination therapies with conventional chemotherapy or radiotherapy will become exceptionally valuable in the future so that more research efforts with SH003 in the field of integrative cancer therapies are much needed. Well-designed RCTs will provide the most accurate estimates of treatment effects while helping physicians to plan the best treatment course for patients. In overall, the potential of SH003 being a chemotherapeutic agent has been well-presented so far; however, continuous further work to facilitate better cancer patient care in the near future.
We appreciate your time and consideration,
Respectfully,
Reviewer 2 Report
Presented topic is interesting. Nevertheless, this manuscript have some serious flag.
Herbal mixture should better describe such as composition, used variants and others. Are used any methods for standardization mixture? In the discussed studies, is the same mixture always used?
Figure 2. Presented effect are not hierarchically equal on dependent on each other.
Table 1, please add used dose of SH003. How physically relevant are the amounts used?
Chapter 4.6. What is the effect of the combination of these substances, possible synergic effect should be also presented and discussed. Please add used concentration/dose. Their synergy with cytostatic should be more discussed.
Minor
Line 99 single dose, or daily dose?
Author Response
Dear Reviewer:
First of all, we would like to express our deepest gratitude for reviewing our manuscript. Please review below for the answers to each comment. All the changes are visible in red from the revised manuscript.
Herbal mixture should better describe such as composition, used variants and others. Are used any methods for standardization mixture? In the discussed studies, is the same mixture always used?
-> SH003 extracts were provided by HANPOONG (HANPOONG PHARM & FOODS Co., Jeonju, Korea), which followed good-manufacturing-practice (GMP) procedures. In brief, Astragalus membranaceus (333 g), Angelica gigas (333 g), and Trichosanthes kirilowii Maximowicz (333 g) were mixed at 1:1:1 ratio and then extracted with 10 times volume of 30% ethanol at 100°C for 3 h. This process was performed 2 times. The extract was dried at reduced pressure (40 Torr) at 60°C for 18 h.
Figure 2. Presented effect are not hierarchically equal on dependent on each other.
-> Please review our updated figure.
Table 1, please add used dose of SH003. How physically relevant are the amounts used?
-> Please review the updated Table 1.
Chapter 4.6. What is the effect of the combination of these substances, possible synergic effect should be also presented and discussed. Please add used concentration/dose. Their synergy with cytostatic should be more discussed.
-> “In brief, a number of our studies has demonstrated that each putative active compound mainly regulates the signaling pathways in apoptosis or autophagy, which are the key anti-cancer targets of SH003. Thus, SH003 is expected to have anti-cancer effects through the synergistic effect of these compounds although this should be proved by non-clinical studies. Moreover, the previous study for chemical profiling of SH003 identified several constituents whereas the anticancer effect of single components and their combination are still unknown [16]. However, there are still undiscovered active components in SH003. Thus, further studies for the identification of new compounds in SH003 and for the investigation of synergistic interactions of multi-components should be performed.”
Line 99 single dose, or daily dose?
-> In brief, rats were orally administrated with SH003 (0, 500, 1,000 and 2,000 mg/kg) every day during 2, 4 and 13 weeks
We appreciate your time and consideration,
Respectfully,
Round 2
Reviewer 2 Report
Manusript was significantly improved.
Only minor points can be taken.
Figure 2 NF-kB is also deeply associated with drug resistance.
Table 2. ref. 47 and 57 used dose (587.569 mg/kg) should not be listed this way.
mg/kg in vitro study?
Author Response
Dear Reviewer:
Thank you for the prompt response, and we appreciate your comments.
We have added "3NF-κB inactivation" in our figure and also deleted "in vitro" in Table 1, ref 58.
We again appreciate your time and help,
Respectfully,